# Hierarchical self-assembly of 3D lattices from polydisperse anisometric colloids

Binbin Luo[1,6], Ahyoung Kim[1,6], John W. Smith[1], Zihao Ou [1], Zixuan Wu[1], Juyeong Kim [1,2,5] & Qian Chen[1,2,3,4]

Colloids are mainly divided into two types defined by size. Micron-scale colloids are widely used as model systems to study phase transitions, while nanoparticles have physicochemical properties unique to their size. Here we study a promising yet underexplored third type: anisometric colloids, which integrate micrometer and nanometer dimensions into the same particle. We show that our prototypical system of anisometric silver plates with a high polydispersity assemble, unexpectedly, into an ordered, three-dimensional lattice. Real-time imaging and interaction modeling elucidate the crucial role of anisometry, which directs hierarchical assembly into secondary building blocks—columns—which are sufficiently monodisperse for further ordering. Ionic strength and plate tip morphology control the shape of the columns, and therefore the final lattice structures (hexagonal versus honeycomb). Our joint experiment–modeling study demonstrates potentials of encoding unconventional assembly in anisometric colloids, which can likely introduce properties and phase behaviors inaccessible to micron- or nanometer-scale colloids.

[1] Department of Materials Science and Engineering, University of Illinois, Urbana, IL 61801, USA. [2] Materials Research Laboratory, University of Illinois, Urbana, IL 61801, USA. [3] Department of Chemistry, University of Illinois, Urbana, IL 61801, USA. [4] Beckman Institute for Advanced Science and Technology, University of Illinois, Urbana, IL 61801, USA. [5] Present address: Department of Chemistry and Research Institute of Natural Sciences, Gyeongsang National University, Jinju 52828, Republic of Korea. [6] These authors contributed equally: Binbin Luo, Ahyoung Kim. Correspondence and requests for materials should be addressed to Q.C. (email: qchen20@illinois.edu)

Anisometry is a potent strategy for biological and artificial building blocks to integrate the merits of both micron- and nanometer-scale dimensions[1–7]. For example, bundling of anisometric tropocollagen helices into the fibrous protein collagen allows multiple building blocks to work in concert under loads, giving collagen its exceptional elastic energy absorption[4]. Nanoscale DNA strands can be engineered to braid into micron-scale filaments to build programmed architectures[6]. Similarly shaped synthetic nanowires can serve as ideal energy transport materials to direct longitudinal conduction of quantum particles such as electrons, phonons, and photons[1,5,8,9]. However, despite extensive work on micron-scale colloids[10–16] and nanoparticles[17–21], anisometric colloids as a promising and emergent particle type are still greatly underexplored. Most work has focused on one-dimensional anisometric building blocks (filaments, tubes, wires, etc.)[1,4–6]. Recent work by our group and others explored shapes like anisometric metallic plates[2,3], which were shown to be great candidates for self-assembly studies. On the one hand, a nanoscale thickness (and therefore low gravitational drag) allows the plates to diffuse freely in solution, which enables their assembly in three dimensions (3D). In comparison, previous work on micron-scale colloids has required either a precise match with solvent density[11,22] or an external field[23] to build 3D structures against gravity. These experimental requirements pose limitations which are in principle irrelevant here (Supplementary Note 1). On the other hand, a micron-scale basal plane makes plates visible under optical microscopy, making it possible to track their translational and rotational motions one-by-one and thus elucidate the assembly dynamics from real time and real space imaging, which was not accessible in nanoplate systems characterized previously by small-angle X-ray scattering (SAXS)[24–27], small-angle neutron scattering (SANS)[26], or electron microscopy with dried samples[28,29]. In comparison, although nanoparticles can assemble into 3D structures[18,30,31], understanding and predictive engineering of their assembly pathways have been challenging due to the technical difficulty in direct imaging of such processes in solution[17,32,33]. However, despite the great advantages from anisometric plates, the observed assembly into one-dimensional lamellar superlattices from these plates is the same behavior seen in nano-sized metallic plates[34,35]; the potential of anisometry to enrich phase behavior was not fully harnessed.

Here we show that the anisometry of silver plates (having a micron-scale basal plane and a nanoscale thickness), when utilized meticulously, opens the design space of colloidal self-assembly to a regime not easily accessible before. Anisometry not only brings the technical ease for studying self-assembly, it also renders strongly directional interactions that favor hierarchical self-assembly along one plate axis at a time. In this way, the initial products in our system, hereafter referred to as "columns", embody the concept of secondary building blocks in molecular crystals[36], which can further assemble into ordered 3D lattices due to a size-focusing effect (an effective narrowing of the building block size distribution). The apparent shape of columns can be predictively manipulated by ionic strength or the extent of plate tip truncation, which leads to distinct final lattice structures. Such unexpected 3D lattices have not been observed in previous systems of metallic nanoplates[2,3,34,35]. Our study demonstrates the unique secondary building block engineering opportunity enabled by anisometry as well as the greatly enriched phase behaviors of anisometric colloids, which opens doors to incorporating their unique mechanical, optical, and electronic properties into 3D assemblies[1,18].

## Results

**Anisometric plates self-assembly in 3D.** Anisometric silver plates studied in this work have a triangular basal plane and tunable extent of tip truncation (Fig. 1a, Supplementary Fig. 1, Supplementary Table 1) that we later show as critical to the selection of final lattice structures. The plates were colloidally synthesized following literature methods of seeded growth with slight modification[2,37]. Tip truncation results from the formation of alternating sides during the kinetic expansion of the basal plane, as explained by Lofton and Sigmund[38] in the silver halide model. The first plate type we studied has a long side length $L$ of $1.49 \pm 0.24$ μm, a short side length $L'$ of $0.79 \pm 0.27$ μm, and a plate thickness $t_{plate}$ of $35 \pm 5$ nm (Fig. 1a, Supplementary Fig. 1). The aspect ratio of the plates, defined as the ratio of circumscribed diameter to the thickness, is as high as ~66. Plates are coated with negatively charged carboxylate–thiols[2] and thus electrostatically repel each other when dispersed in deionized water (Fig. 1a, Supplementary Note 2). The plates exhibit vivid in-plane and out-of-plane rotations in 3D as observed under optical microscopy, with a vertical fluctuation height[39] as much as 11 times the plate thickness (Supplementary Fig. 2a, b, Supplementary Note 1). Such out-of-plane rotation in 3D, given one order of magnitude difference between the densities of silver and water, is possible due to the nanoscale thickness.

Self-assembly of the dispersed plates was triggered by increasing the ionic strength (0.5 mM NaCl), leading to the formation of standing column secondary building blocks by face-to-face plate stacking (Fig. 1b, Supplementary Fig. 2c, Supplementary Movie 1). The column geometry was verified under optical microscopy, as rotating columns exhibit light, rectangular projections while lying on their sides and dark, polygonal projections while standing vertically on the substrate (Fig. 1c, Supplementary Movie 2). At this ionic strength, van der Waals attraction ($E_{vdW}$) between adjacent plates overcomes screened electrostatic repulsion ($E_{el}$), and because the basal plane has a ~63 times larger interacting area than the plate side, face-to-face stacking into columns is favored over lateral assembly. Specifically, based on the pairwise interaction ($E_{tot} = E_{vdW} + E_{el}$) we computed following a discretized model[40,41], the face-to-face stacking as observed in experiments has an $E_{tot}$ minimum of −6.1 $k_B T$ (Fig. 1d, Supplementary Fig. 3, Supplementary Table 2), while the side-by-side configuration is not as stable, with a negligible $E_{tot}$ minimum of $-0.02$ $k_B T$. Face-to-face stacking of plates within the columns was further verified by SAXS, where the measured face-to-face spacing between adjacent plates, the $d$ spacing, is consistent with that predicted by both the analytical and discretized models for plate–plate interactions (Supplementary Fig. 4, Supplementary Table 3). Based on the $d$ spacing, one column is composed of ~15 plates (Supplementary Fig. 2d, e). The column structure was also observed in our earlier work[2] on plates that have a smaller basal plane area (0.90 μm², versus 3.27 μm² here). The key difference is that, due to their larger basal plane area, the columns in this work experience enhanced van der Waals attraction with the substrate and larger gravitational force, so the columns prefer to stand vertically on the substrate (Supplementary Fig. 2c, Supplementary Note 3), facilitating the subsequent lateral assembly of columns as secondary building blocks.

**Ordering from polydisperse plates via size-focusing.** The standing columns further assemble as their concentration increases into an ordered hexagonal lattice due to a size-focusing effect in the columns (Supplementary Movie 3). This size-focusing effect is a direct result of the plate anisometry. Figure 1e, f summarizes the overall trend as we tilt the microscopy chamber slightly to concentrate the columns. The fast Fourier transform of the final equilibrated structure (FFT, Fig. 1e, inset) reveals a global hexagonal order. Our tracking of adjacent columns in the

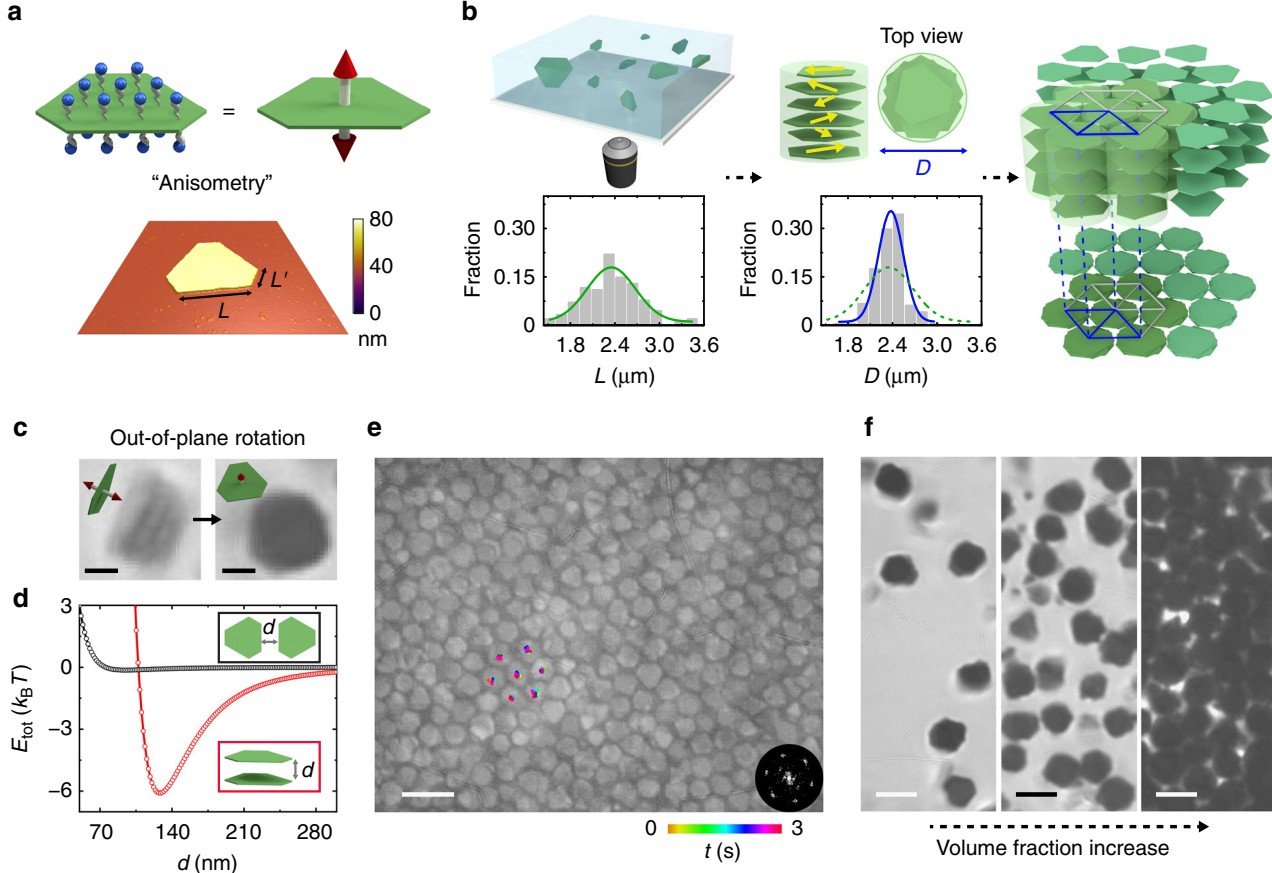

**Fig. 1** Hierarchical self-assembly of polydisperse, anisometric plates into a 3D lattice. **a** Top: schematics of anisometric silver plates (green) coated with thiolated COO⁻ ligands (blue) and their directional attractions (red arrows). Bottom: a representative atomic force microscopy (AFM) scan over an anisometric plate. The color bar represents height. Scan area: 5 μm × 5 μm. **b** Schematics describing the hierarchical building process: plates first stack into columns (yellow arrows denoting the random orientations of the plates), which then assemble into a hexagonal lattice. The graphs below show the distribution of the long side length $L$ of plates (left, green curve) and that of column projection diameter $D$ (right, blue curve). The $L$ distribution was rescaled so that $\bar{L}$ and $\bar{D}$ have the same value (2.36 μm) for direct comparison. **c** Time-lapse optical microscopy images and schematics showing the lying and standing orientations of the same rotating column. **d** Theoretical calculations of the net pairwise interactions $E_{tot}$ of two plates in the face-to-face (red circles) and side-by-side (black circles) configurations as a function of the plate–plate distance $d$ (labeled in inset schematic). **e** An optical microscopy image showing a hexagonal lattice, overlaid with tracked trajectories of the central positions of the column projections. The image was processed as detailed in Supplementary Fig. 5 and Supplementary Note 4. The bottom right inset is a fast Fourier transform (FFT) pattern of the image. **f** Optical microscopy images showing column arrangements as their concentration increases (from left to right). Ionic strength in **c**–**f**: 0.5 mM. The schematics were not drawn to scale. Scale bars: 1 μm in **c**; 5 μm in **e**; 3 μm in **f**

lattice (traces labeled in Fig. 1e) shows highly localized vibrations at the lattice sites, validating the lattice stability. Two details are noted in the hexagonal lattice formation. First, the plates are polydisperse (about 16% based on $L$, 34% based on $L'$) because the growth of micron-sized particles from nano-sized seeds is highly susceptible to local fluctuations in solution, as reported previously[3,37]. This polydispersity is usually considered too high for forming an ordered crystal (for example, <12% polydispersity is required for spheres to crystallize)[42]. However, the secondary building blocks of columns average their sizes due to plate stacking, thereby rendering them greatly reduced polydispersity (8%) to assemble into an ordered lattice (Fig. 1b). Such a size-focusing effect can potentially account for the formation of similar 3D lattices from polydisperse Al(OH)₃ nanoplates presented in earlier work, where ensemble structure was characterized by SAXS[24,25,27]. This effect which enables ordered assembly from polydisperse building blocks is also reminiscent of the monodisperse supraspheres formed from polydisperse CdSe nanoparticles[43], but was achieved here via hierarchical averaging, not self-limiting clustering which requires an intricate balance of

competing forces[43]. Second, the columns are otherwise dispersed at low or intermediate column concentrations (Fig. 1f, Supplementary Fig. 6, Supplementary Movie 4), indicating a weak or negligible attraction between the columns at this ionic strength. The weak attraction is further verified as the columns were observed to diffuse close into contact and then apart due to the absence of strong attraction (Supplementary Fig. 6a, b), such that they only crystallize into ordered lattices when their concentration is sufficiently high[44]. The ordered hexagonal lattices are observed in multiple plate samples of different sizes (Supplementary Fig. 7), further demonstrating the robustness of the 3D hierarchical assembly. The 3D nature of the assembled structures is also verified by scanning electron microscopy (SEM) imaging (Supplementary Note 5), where layers of plates are clearly shown to stack into laterally interacting columns (Supplementary Fig. 8).

A close investigation into the relative rotations of plates in the columns further reveals how the effective shape of columns is reduced into monodisperse, radially symmetric circular columns. As shown in Fig. 2a and Supplementary Figs. 9 and 10, the constituent plates in a column rotate relative to each other

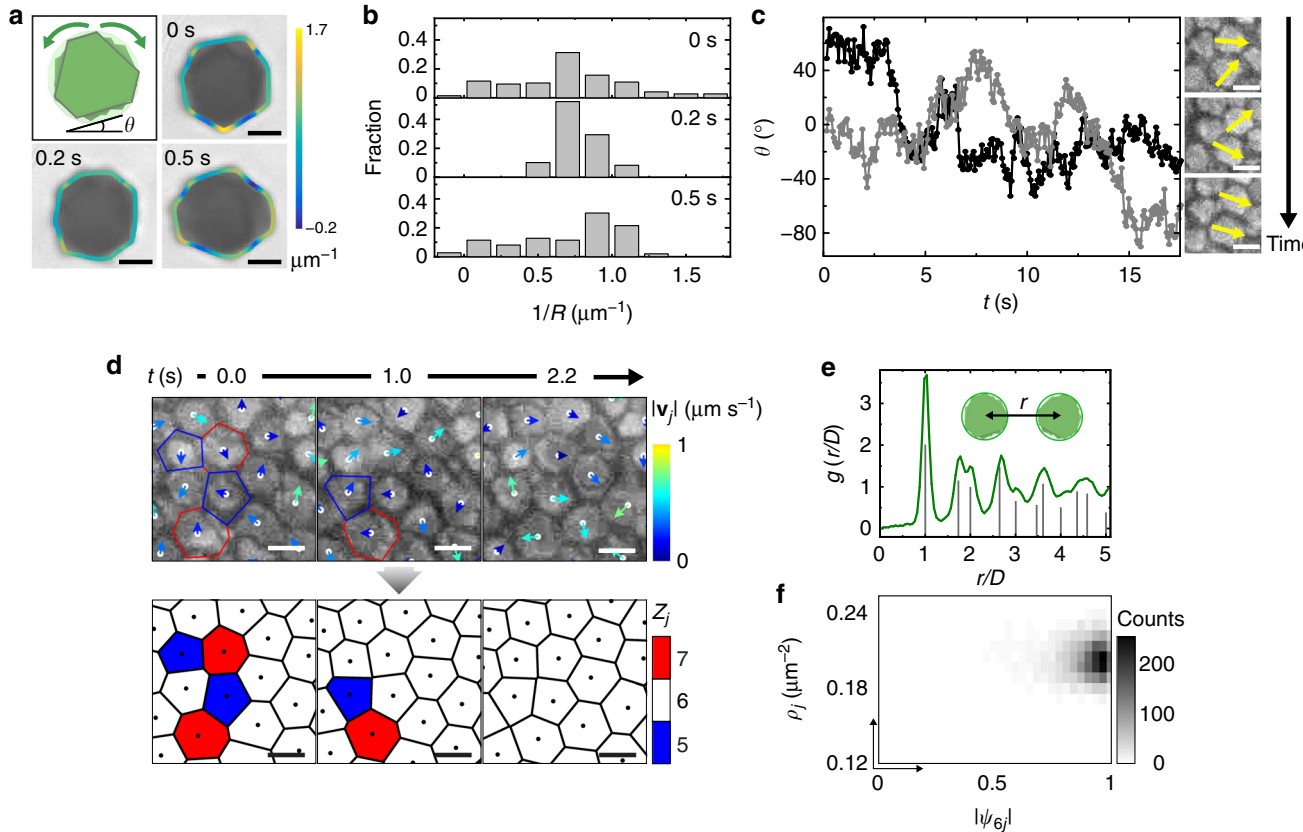

**Fig. 2** Radially symmetric columns assembled from triangular plates. **a** Schematic and time-lapse optical microscopy images of the projections of the same column, overlaid with contour lines color-coded according to the local curvature (the inverse of the locally fitted circle radius $R$)[40]. The color bar represents curvature values. The first-layer plate's orientation $\theta$ is defined as the angle of one long side of the plate relative to the horizontal axis. **b** The distribution of local curvature of the projection contours tracked at 0, 0.2, and 0.5 s in **a**. **c** Orientations of two plates (one in gray, the other in black) in a pair of nearest neighbor columns inside the lattice over time, and time-lapse optical microscopy images with the plate orientations labeled in yellow. The measurement errors are ±5°. **d** Time-lapse optical microscopy images (top) and corresponding Voronoi cell representations (bottom) of the hexagonal lattice, showing the annealing of imperfectly coordinated sites. The arrows in the top panel are color-coded by the velocity magnitude $|\mathbf{v}_j|$ of columns calculated over a period of 0.7 s (12 frames). The color of each Voronoi cell denotes the number of nearest neighbors $Z_j$ per column $j$. **e** Radial distribution function of the lattice in experiment (green curve) and that of an ideal hexagonal lattice (gray lines). The inset defines the center-to-center distance $r$ between columns. A $D$ value of 2.36 µm is used in the plot. **f** The local order–local density ($|\psi_{6j}|, \rho_j$) histogram based on single column tracking of the hexagonal lattice (32 frames in Supplementary Movie 3). Here $|\psi_{6j}| = |\frac{1}{Z_j}\sum_{k=1}^{Z_j} \exp(6i\beta_{jk})|$, where the summation goes over all the nearest neighbors of column $j$, and $\beta_{jk}$ is the angle between the bond linking column $j$ and its $k$th neighbor and an arbitrary reference axis[46,47]. Ionic strength: 0.5 mM. Scale bars: 1 µm in **a**; 2 µm in **c**, **d**

(Supplementary Movie 5) so that the projections of columns dynamically vary. Plates at the first few layers in the column are clearly identified based on their contrast difference under optical microscopy (Supplementary Movie 5). The outlined contours of the column projections were color-coded according to the local curvature (Supplementary Note 6), which fluctuate to average the triangular plates of different orientations into a circular column projection as quantified in Fig. 2b. As a result, the effective circular columns pack closely into a hexagonal lattice when concentrated (Fig. 1b, e), much like how circular disks crystallize at high disk concentrations[44]. The column projection diameter $D$ was obtained from the center-to-center distance between nearest neighbor columns in the lattice. The measured value is 2.36 ± 0.19 µm, which is close to the diameter of a circle circumscribing an average plate (2.32 µm), indicating that the columns in the lattice are closely packed. The plates in a nearest pair of columns are expected to stay "enveloped" by the circular rims, experiencing little physical contact as they rotate. Indeed, the orientations of two such plates span a range over 120° over time as shown in Fig. 2c and Supplementary Movie 6, indicating a full 360° rotation range given the three-fold rotational symmetry of the plate. The

Pearson correlation coefficient[45] of the plate orientations ($\theta_1$, $\theta_2$) sampled for $n$ continuous frames, defined as $\frac{\sum \theta_{1f}\theta_{2f} - n\overline{\theta_1}\,\overline{\theta_2}}{\sqrt{(\sum \theta_{1f}^2 - n\overline{\theta_1}^2)}\sqrt{(\sum \theta_{2f}^2 - n\overline{\theta_2}^2)}}$, is as low as −0.01 ($f$ being the frame number; $n = 300$), confirming a weak angular correlation between plates in adjacent columns.

The tracked column positions and their local motions at the lattice sites reveal not only detailed lattice relaxation dynamics but also a series of parameters characteristic of a hexagonal lattice (Supplementary Note 7, Supplementary Fig. 11). For example, we mapped out the velocity $\mathbf{v}_j$ for each column $j$ (top panel, Fig. 2d) and the Voronoi cell representation of the lattice (bottom panel, Fig. 2d) defining areas specific to each column. This analysis shows the dynamic evolution of imperfectly arranged columns as the number of nearest neighbors ($Z_j$) reaches to 6 of a perfect hexagonal lattice (Fig. 2d). The radial distribution function[12] $g(r)$ of the column lattice is similar to that of an ideal hexagonal lattice with persisting peaks (Fig. 2e). The calculated six-fold local bond orientation order parameter[46,47] per column $j$, $|\psi_{6j}|$, together with the computed local density $\rho_j$, shows that the columns are both of

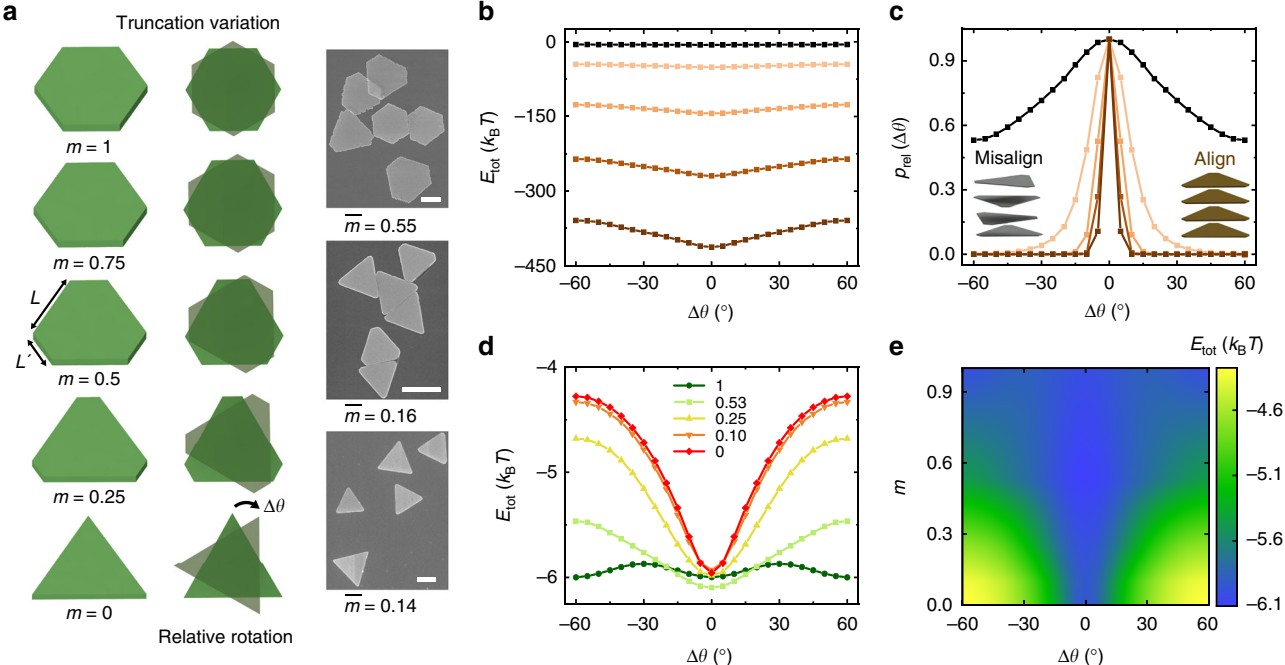

**Fig. 3** The effective shape of columns modulated by ionic strength or plate tip truncation. **a** Schematics (left) and SEM images (right) showing plates of a systematically varying extent of truncation $m$ (defined as $L'/L$). The top right SEM image shows the plates used in the hexagonal lattice. The corresponding truncation distributions are shown in Supplementary Fig. 13. Scale bars: 1 μm. **b** A graph showing how the computed pairwise interaction $E_{tot}$ between two stacked plates ($m = \overline{L'}/\overline{L} = 0.53$, corresponding to the plates used for hexagonal lattice) changes as a function of $\Delta\theta$ at different ionic strengths (from top to bottom: 0.5, 1, 1.5, 2, and 2.5 mM, respectively). Note that 0.5 mM is used in the hexagonal lattice assembly. **c** Relative probability distributions of $\Delta\theta$ based on a Boltzmann distribution argument at different ionic strengths, computed from the interaction energy plot in **b** (color coded the same as in **b**). The inset schematics show misaligned plate orientations inside column (left) at low ionic strength (0.5 mM) and well-aligned plate orientations (right) at high ionic strength (2.5 mM). **d** A graph showing how the computed pairwise interaction $E_{tot}$ between two stacked plates changes as a function of $\Delta\theta$ at different extent of truncation $m$ as labeled. Here the calculations are conducted at a fixed basal plane area (3.27 μm²). Ionic strength: 0.5 mM. **e** Pairwise interaction strength as a function of $\Delta\theta$ and truncation $m$ at their energy minimum spacing $d$. Ionic strength: 0.5 mM

a high density $\rho_j$ and a high local order $|\psi_{6j}|$ (83% of all the values larger than 0.8 and an average $|\psi_{6j}|$ value of 0.87 over all the columns) (Fig. 2f, Supplementary Fig. 11e, f), indicating the high crystallinity of hexagonal lattice from column secondary building blocks[46].

**Modulating the effective shape of columns.** Having elucidated that the circular shape of columns leads to hexagonal lattices, we next demonstrated that the effective shape of columns can be controlled by changing the ionic strength or the extent of plate tip truncation, potentially leading to different lattice structures. Here we define a parameter $m$ as the ratio between the short ($L'$) and the long ($L$) side lengths of a plate, to quantify the extent of tip truncation. As shown in both the schematics and SEM images (Fig. 3a), an $m$ value of 0 corresponds to a triangular plate with sharp tips and 1 to an equilateral hexagonal plate. Regarding the ionic strength effect, given the same type of plates as studied above (the plates that are assembled into a hexagonal lattice), we calculated the interactions $E_{tot}$ between a stacked plate pair with varying relative in-plane orientations $\Delta\theta$ at their energy minimum spacings, as a function of ionic strength (Fig. 3a, b, see details in Methods). At a relatively low ionic strength (0.5 mM, the experimental condition for forming hexagonal lattice), $E_{tot}$ is weakly dependent on $\Delta\theta$ over a range of −60° to 60°. Based on a Boltzmann distribution argument (relative probability, $p_{rel}(\Delta\theta) \approx e^{-\Delta E/k_B T}$, where $\Delta E$ is the $E_{tot}$ difference for two plates at different relative orientations, see Fig. 3c), large plate–plate relative rotations are allowed with low energy penalty. This prediction is also consistent with our experimental observations of fast relative

rotations of plates at this ionic strength condition (Fig. 2a, b). The interaction profile becomes increasingly steeper with respect to plate relative rotation as ionic strength increases, restraining the relative rotations between adjacent plates so that the plates within the columns favor aligned orientations (Fig. 3c). We verified this trend in experiments at high ionic strength (2.0 mM). At this ionic strength, the electrostatic repulsion is more screened, leading to a decreased separation between plates in the columns, higher enthalpic penalties for plate relative rotation, and therefore dampening of plate relative rotations as well as a significant increase in the lateral attraction between columns. Irregular, fractal-like assemblies from columns were observed from the same plate sample at high ionic strength (2.0 mM; Supplementary Fig. 12). In addition, plate tip truncation also drastically changes the allowed relative rotations of plates and thus the effective shape of columns. For pairwise interaction $E_{tot}$ calculated at a constant ionic strength and a constant basal plane area, but at varying $m$, we find a more hexagonal shape (a larger $m$) renders a flatter interaction profile over different $\Delta\theta$ values (Fig. 3d, e). The plates with a more triangular shape (a smaller $m$), on the other hand, have a steeper potential well disfavoring relative rotations between plates. This interaction profile, on the column level, maintains the generic three-fold symmetry of triangular plates instead of a smeared-out circular shape. This detail of tip truncation was not recognized in the previous literature[2], due to the lack of importance in inter-column interactions in those systems.

**Experimental realization of 3D honeycomb lattice.** Finally, we experimentally realized the triangular columns predicted above, and in doing so observed a different final assembly: honeycomb,

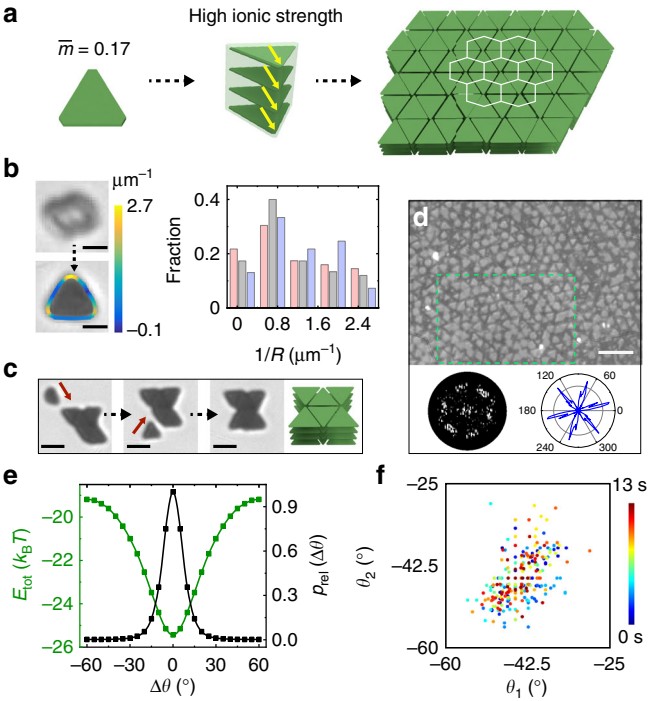

**Fig. 4** Experimental observation of plates assembling into 3D honeycomb lattice domains. **a** Schematics showing the hierarchical self-assembly process. The plates here have the dimensions, $L$: 1.28 ± 0.27 μm; $L'$: 0.19 ± 0.19 μm; $t_{plate}$: 22 ± 2 nm; $\bar{m} = 0.17$. The yellow arrows denote the aligned plate orientations in the column. **b** Time-lapse optical microscopy images (left) showing the lying and standing orientations of the same rotating column. The bottom image is overlaid with contours color-coded according to local curvature. The color bar represents curvature values. The right plot shows the local curvature distributions of the standing column projection contours over time (pink: 0 s; gray: 1 s; purple: 1.5 s). Ionic strength: 3.2 mM. **c** Time-lapse optical microscopy images and schematic showing the assembly process of the columns into a "hexamer". **d** An optical microscopy image of the assembled lattice with multiple ordered honeycomb domains. The image was processed following details in Supplementary Fig. 5. The bottom left image is an FFT of one honeycomb lattice domain in the green box. The bottom right plot is an orientation distribution map of the plates in the same lattice domain. **e** Pairwise interaction (green curve) and relative probability distribution normalized to $\Delta\theta = 0°$ (black curve) of the two stacked plates as a function of plate relative orientation $\Delta\theta$. **f** A scatter plot showing the orientations of two plates in a nearest neighbor pair in the honeycomb domain over time. Each data point color corresponds to a different time. The measurement errors are ±2°. The color bar represents time. Ionic strengths in **c**–**f**: 3.5 mM. Scale bars: 1 μm in **b**; 2 μm in **c**; 5 μm in **d**

not hexagonal, lattice domains are formed upon decreasing the extent of plate tip truncation and increasing the ionic strength (Fig. 4a). We synthesized less truncated triangular plates while maintaining anisometry ($\bar{m} = 0.17$, Fig. 4a, Supplementary Fig. 14). As shown in Fig. 4b, these plates assemble into columns at increased ionic strengths (3.2–3.5 mM). The column projections are mostly triangular, with the contoured local curvature distributions staying roughly constant over time (Fig. 4b, Supplementary Movie 5). Such distributions suggest that the stacked plates within the columns remain aligned, with negligible relative plate rotation. This observation is consistent with our interaction modeling (Fig. 4e), which shows that even a small relative plate rotation of 20° can be strongly disfavored due to an interaction energy penalty as high as 3 $k_B T$. The disfavoring of plate

misalignment in the columns leads to a triangular column projection, which encodes an assembly symmetry distinct from the isotropic, circular ones studied above. At low column concentration, the triangular columns assemble side-by-side into small clusters via individual plate or column addition (see "hexamer" formation in Fig. 4c), indicating a directional attraction between columns that is strong enough to hold them together once coming close. At high column concentration, the elementary side-by-side attachment extends to 3D honeycomb lattice domains (Fig. 4a, d with inset FFT).

In comparison with the previous hexagonal lattice, the honeycomb lattice domains exhibit high orientational order and positively correlated plate rotation in a nearest neighbor pair of columns. The first-layer plate orientation distributions (by measuring the orientations of the three sides of a plate; Supplementary Note 8) show a six-fold orientational order (bottom right plot in Fig. 4d), consistent with a honeycomb lattice. The two plates in a nearest neighbor column pair have greatly hindered rotations that are positively correlated (a Pearson correlation coefficient of 0.51, Fig. 4f, Supplementary Movie 6), which suggests side-by-side arrangements. Furthermore, upon an ionic strength decrease from 6 to 2 mM in a similarly truncated plate system ($\bar{m} = 0.16$), we observed a transition from the space-filling honeycomb lattice composed of triangular columns to a loosely packed structure which exhibits hexagonal packing due to increased electrostatic repulsion and lowered penalty for relative plate rotation (Supplementary Fig. 15). Such precise control of the final 3D lattice structures arises from the effective shape and consequent coordination symmetry of the column secondary building blocks. Anisometry in the plates drives 3D hierarchical assembly, which proceeds one step at a time and enables engineering of the intermediate product, a strategy molecular crystal engineering adopts to obtain a rich library of crystal structures[36]. Control experiments on the 2D assembly of silver plates at different plate concentrations did not show ordered structure formation (Supplementary Fig. 16).

## Discussion
In our study of anisometric colloidal self-assembly, two otherwise separated length scales merge to induce 3D hierarchical self-assembly. Fundamentally the anisometry encodes highly directional interactions and induces the formation of secondary building blocks, which enable the selection of final structures from polymorphs over an otherwise complicated free energy landscape. Beyond the conventional small molecular ligands used here, macromolecular shelling[48] or ligands that are responsive to external fields[49,50] can also be introduced to the anisometric colloids to render patchy interactions or non-equilibrium assemblies, thereby aiding self-assembly into further complicated 3D superstructures. The colloidal shapes as well as the basic quantum-confinement properties can have rich variations. For example, star-shaped or other polygonal[51] basal planes have architecture-dependent plasmonic coupling which can be potentially modulated in the form of 3D arrays as optoelectronic devices[52]. Remarkably, our experimentally observed 3D hierarchical lattices are different from those adopted by simulated triangular prisms, which only form into honeycomb lattices due to hard-core exclusion[53], yet resembling those simulated for high aspect-ratio circular disks[54,55]. The combination of positional order and orientational disorder we observed is reminiscent of molecular systems[56,57]. For example, plate-like molecules can stack into columnar phases[56] to harness longitudinally-favored charge conduction paths and show greater conductivity along the columns than in the perpendicular direction[58]. Molecules such as boron-containing "connectors" or hydrocarbon "linkers" can

form into covalent organic frameworks, with cage architecture and rotational order developed upon a delicate balance between intermolecular attraction (van der Waals, π–π stacking) and electrostatic repulsion[57]. The formation of covalent organic frameworks is similar to the formation of our honeycomb lattice from triangular plates, upon a balance of interparticle interactions. The real-time dynamics observed here can potentially provide insight into phenomena taking place in those molecular systems.

## Methods

**Chemicals**. Sodium citrate tribasic dihydrate (≥99.0%, HOC(COONa) $(CH_2COONa)_2 \cdot 2H_2O$, Sigma-Aldrich), sodium borohydride (99%, $NaBH_4$; Sigma-Aldrich), silver nitrate (≥99.0%, $AgNO_3$, Sigma-Aldrich), L-ascorbic acid (BioXtra, ≥99.0%; Sigma-Aldrich), citric acid monohydrate (99.0–102.0%, $(HO_2CCH_2)_2$-C $(OH)(CO_2H) \cdot H_2O$; Fisher Scientific), sodium chloride (99.3%, NaCl; Fisher Scientific), hydrogen peroxide (30 wt %, $H_2O_2$; Macron), and 2-(2-[2-(11-mercapto-undecyloxy)-ethoxy]-ethoxy)-ethoxy-ethoxy-ethoxy-ethoxy-ethoxy-acetic acid (≥95%, HS $(CH_2)_{11}(OC_2H_4)_6OCH_2COOH$; Prochimia Surfaces) were purchased and used without further purification. All glassware was treated with aqua regia (prepared by mixing HCl and $HNO_3$ with a volume ratio of 3:1), thoroughly rinsed with water, and dried immediately before use. Nanopure water (with a resistivity of 18.2 MΩ cm at 25 °C) purified by a Milli-Q Advantage A10 system was used for all washing and solution preparation.

**Synthesis of anisometric silver plates**. The silver plates of controlled extent of truncation were synthesized following a slightly modified seeded-growth method[2,37]. Silver seeds were prepared by adding 25 mL $AgNO_3$ (0.1 mM), 600 μL sodium citrate (75 mM), and 60 μL hydrogen peroxide (30 wt %) in sequence to a 125 mL Erlenmeyer flask stirring at 300 r.p.m. at room temperature. Next, the stirring speed was increased to 1050 r.p.m. before rapid injection of 250 μL freshly prepared aqueous $NaBH_4$ solution (0.1 M). This $NaBH_4$ solution was prepared by adding solid $NaBH_4$ to water, which had been precooled in an ice bath for ~10 min and then vortexing immediately. The obtained light yellow solution turned blue about 4–10 min after the injection of the $NaBH_4$ solution and was kept stirring for another 5 min after this color change. Then the mixture was centrifuged (Eppendorf Centrifuge 5804) at 10,500 r.p.m. for 8 min and most of the supernatant was carefully removed, leaving ~50 μL sediment. This sediment was redispersed in 10 mL 0.94 mM sodium citrate solution as the seed solution. A UV–Vis spectrum of the as-prepared seeds was measured using a Scinco S-4100 PDA spectrophotometer. In different batches of seed preparation, the extinction peak positions were at 700–800 nm. Based on the UV–Vis spectrum, we diluted the seed solution using 0.94 mM sodium citrate solution to a concentration with an extinction intensity of 0.012 or 0.0024 at the maximum peak position; this diluted solution was later used as our seed solution stock to grow silver plates.

All the parameters used in silver plate growth (different size and different extent of truncation $m$) are listed in Supplementary Table 1. Below are the synthesis details of the two typical types of plates for which we studied self-assembly behavior. For plates with $\bar{m} = 0.55$ (used in Figs. 1 and 2), 0.75 mL of L-ascorbic acid (0.1 M) was added to 20 mL of the seed solution stock kept in a 40 mL glass vial (Fisherbrand, 29 mm × 94 mm) as growth solution I. At the same time, a separate growth solution II was prepared by mixing 20 mL $AgNO_3$ (1.0 mM), 125 μL citric acid (0.1 M), and 100 μL sodium citrate (1.5 mM). Growth solution II was added to growth solution I dropwise through a syringe pump (Fusion 200, Chemyx Inc.) at a rate of 0.2 mL min⁻¹ to start the plate growth. During the growth process, the growth solution was shaken vigorously at room temperature at a speed of 700 r.p.m. for the first cycle and 1000 r.p.m. for the later cycles (Eppendorf, ThermoMixer C). After every 10 min of injection, which is one growth cycle, two-thirds of the growth solution in the vial was removed, and the remaining solution was used for the next growth cycle to grow larger plates. The solution removed after growth cycle 5 was collected and stored in 8 mL glass vials (Wheaton, borosilicate glass, 17 mm × 60 mm) for further thiol modification. For plates with $\bar{m} = 0.17$ (used in Fig. 4), all the procedures and reactant concentrations are the same as above, except that (i) during the plate growth, the growth solution was kept stirring vigorously at a speed of 900 r.p.m. for the first cycle and 800 r.p.m. for the later cycles; (ii) the injection rate of growth solution II was 0.4 mL min⁻¹, with 5 min injection as one growth cycle; and (iii) plates after growth cycle 3 were collected and used for thiol modification.

**Thiol modification of silver plates**. The silver plates obtained above were coated with citrate ligands. These ligands were exchanged with carboxylate–thiols (HS $(CH_2)_{11}(OC_2H_4)_6OCH_2COOH$) to improve colloidal stability in solution. Specifically, 100–200 μL, 7.93 mM thiol solution was added to 2 mL silver plate solution collected from the growth cycle. This 7.93 mM thiol solution was prepared by dissolving 5 μL thiol as received from Prochimia in 1 mL water. The reaction solution was then allowed to sit overnight without stirring to complete the ligand exchange. After that, the plates with exchanged ligands sediment to a thin layer at

the bottom of the vial. The supernatant was carefully removed, and 2 mL water was added to the sediment. After 5 h of additional sedimentation, the supernatant was removed and the final sediment product was stored in water in an 8 mL glass vial.

**Plate characterization**. The shape and size distribution of silver plates and the assembled 3D lattice structures were characterized using SEM (Hitachi S4800). The plate thickness was measured using either tapping-mode AFM (Asylum Research Cypher) or SEM when the plates were vertical to the viewing angle. The silicon wafer used in the AFM measurement was first sonicated with acetone and isopropanol each for 15 min, and then treated with oxygen plasma (Harrick Plasma Cleaner PDC-23G) at medium power for 1 min to render the wafer surface hydrophilic. Then one drop of silver plate solution (~5 μL) was pipetted on the wafer and allowed to dry under ambient conditions before the AFM measurements. The 3D lattice structures formed in solution in the optical microscopy chamber were controllably dried in the chamber under ambient conditions, to maximally maintain the structure for SEM imaging.

**Optical microscopy imaging**. Approximately 150 μL silver plate suspension after thiol modification was added to an eight-well chamber (composed of a 0.13–0.17-mm-thick cover slip with plastic walls, Chambered Coverglass, Thermo Scientific). Bright-field optical microscopy imaging was performed using a Zeiss inverted microscope (Axiovert 200) with a ×100 oil objective (NA = 1.45) and ×1 or ×1.6 post magnification. Optical microscopy movies were recorded using a complementary metal-oxide-semiconductor camera (Edmund Optics 5012 M GigE) at a rate of 17.2 or 21.8 frames per second (fps).

**SAXS measurements**. SAXS measurements were conducted on the truncated plate sample ($\bar{m} = 0.55$) to confirm that plates stack face-to-face into columns and to measure the equilibrium plate-to-plate spacing between adjacent plates in the columns at certain ionic strengths. Specifically, the columns were prepared by adding salt to a concentrated plate suspension to an ionic strength that induces self-assembly (0.9 mM). Next, SAXS samples were prepared by gently transferring 100 μL as-prepared columns at the desired ionic strength to a quartz capillary (1.5 mm outer diameter, 0.01 mm wall thickness; Hampton Research), which was then sealed with poly(tetrafluoroethylene) tape to prevent any evaporation. We waited for 3 h for the sample to equilibrate at the bottom of the capillary. Right before the SAXS measurement, the capillary was gently shaken a few times to disperse the concentrated columns uniformly at the bottom of the capillary. Transmission SAXS spectra were collected for 10–30 min using a home-built setup (Forvis Technologies, Santa Barbara) with a Xenocs GeniX3D Cu Kα ultralow divergence X-ray source (1.54 Å/8 keV) with a divergence of ~1.3 mrad and a Pilatus 300 K 20 Hz hybrid pixel detector (Dectris). FIT2D (European Synchrotron Radiation facility, http://www.esrf.eu/computing/scientific/FIT2D) was used to analyze the two-dimensional (2D) scattering plots and obtain the intensity ($I$) and scattering vector ($q$) profile (Supplementary Fig. 4). The fact that $q$ values in the SAXS spectra are integer multiples indicates a 1D lamellar structure[2] (see Supplementary Table 3). Due to the large plate-to-plate spacing of the assembled chains, the first two peaks (those with $q$ values <0.01 Å⁻¹) were overwhelmed by the profile of the unscattered beam. Consequently, the $q$ values of the first two peaks were derived from higher-order peaks (Supplementary Table 3). The equilibrium center-to-center distance between neighboring plates in the columns, $d_c$, was calculated based on the derived first-order peak position ($q_1$) as $d_c = \frac{2\pi}{q_1}$.

**Analytical modeling of pairwise interactions**. In Fig. 1d, the pairwise interaction $E_{tot}(d)$ was evaluated as a sum of van der Waals attraction[35] between silver in water and electrostatic repulsion between charged ligands[34], $E_{tot}(d) = E_{vdW}(d) + E_{el}(d)$, where

$$E_{vdW}(d) = -\frac{H_{eff}S}{12\pi}\left[\frac{1}{(d)^2} - \frac{2}{(d+t_{plate})^2} + \frac{1}{(d+2t_{plate})^2}\right] \quad (1)$$

and

$$E_{el}(d) = \frac{\varepsilon\varepsilon_0\zeta^2 S}{b}\left[1 - \tanh\left(\frac{d-2t_{ligand}}{2b}\right)\right]. \quad (2)$$

Here, $d$ is the face-to-face spacing between two neighboring plates, $t_{plate}$ is the plate thickness, and $S$ is the plate basal plane area. In the electrostatic potential, $\varepsilon$ is the relative permittivity of water, $\varepsilon_0$ is the vacuum permittivity, $\zeta$ is the zeta potential of the thiol-coated silver plates in water, and $b$ is the Debye length, given (in nanometers) for an aqueous solution of sodium chloride salt as $b = \frac{0.304}{\sqrt{I}}$ for ionic strength $I$ (in moles per liter). In addition, $t_{ligand}$ is the thickness of the carboxylate–thiol monolayer on the plate surface[34]. Here we assumed that the two plates are in a perfectly aligned, face-to-face configuration.

Because the plates in this system achieve large interparticle separations, the Hamaker constant was treated as an "effective" Hamaker constant in such a way that the retardation of the van der Waals attraction at large separations is

accounted for. In other words,

$$H_{\text{eff}} = \frac{H_0}{1 + \frac{ad}{\lambda}}. \tag{3}$$

Here $a$ is constant describing the geometry of the interaction ($a = 5.3$ for two surfaces, $a = 11$ for two spheres, and $a = 14$ for a sphere and a surface) and $\lambda$ is a parameter describing the onset length scale for retardation. $H_0$ is the Hamaker constant for silver and silver interacting across water at zero separation ($4.0 \times 10^{-19}$ J)[59,60]. This model was previously shown to be a simple but robust approximation for more complicated retardation theories[41,61]. The value of $\lambda$ can be approximated by

$$\lambda = \frac{hc}{\Delta E}, \tag{4}$$

where $h$ is Planck's constant, $c$ is the speed of light, and $\Delta E$ is a "characteristic energy" in the electronic spectrum of the material, commonly taken to be the first ionization energy. For silver, $\lambda$ is therefore about 164 nm.

Net interaction potentials and the locations of their secondary minima at different ionic strength conditions are indicated in Supplementary Fig. 3, as calculated using the values in Supplementary Table 2.

**Discretized modeling of pairwise interactions.** The interaction models discussed above do not account for the relative rotation of the plates perpendicular to the column axis. Therefore, we also constructed a coarse-grained (CG), pairwise interaction model to evaluate the effect of relative plate rotation on the interaction potential (Figs. 3 and 4). Anisometric plates were discretized as $N = 21,392$ to $26,744$ beads per plate (depending on the extent of truncation) with beads 17.5 nm in diameter (Supplementary Fig. 3a). Beads were treated as point-like particles, whose van der Waals and electrostatic interactions can be described as

$$e_{\text{vdW}}(r_{\text{bead–bead}}) = \frac{-H_{\text{eff}}\sigma^6}{\pi^2 r_{\text{bead–bead}}^6} \tag{5}$$

and

$$e_{\text{el}}(r_{\text{bead–bead}}) = \frac{Z_{\text{bead}}^2 e^2}{4\pi\varepsilon\varepsilon_0 r_{\text{bead–bead}}} e^{-r_{\text{bead–bead}}/b}, \tag{6}$$

where $r_{\text{bead–bead}}$ is the distance between beads in a pair, $\sigma$ is the bead size, and the remaining variables have the same meaning as in the above analytical modeling[40,41]. The charge of a bead, $Z_{\text{bead}}$, was approximated first by calculating the surface charge density, $\rho_{\text{surf}}$, based on the zeta potential, using the Grahame equation:[62]

$$\rho_{\text{surf}} = \sqrt{8I\varepsilon\varepsilon_0 RT} \sinh\left(\frac{e\zeta}{2k_{\text{B}}T}\right) \tag{7}$$

finding the total charge of the plate surface, and then dividing this charge across all $N$ beads.

The van der Waals interaction between anisometric plates was taken as the sum of all bead–bead van der Waals interactions, i.e.,

$$E_{\text{vdW}}^{\text{CG}}(d) = \sum_{i=1}^{N}\sum_{j=1}^{N} e_{\text{vdW}}(r_{\text{bead–bead}}) = \sum_{i=1}^{N}\sum_{j=1}^{N} \frac{-H_{\text{eff}}\sigma^6}{\pi^2 \left|\mathbf{r}_i - \mathbf{r}_j\right|^6} \tag{8}$$

for beads $i$ in one plate and beads $j$ in the other. Similarly, the plate–plate electrostatic interaction was evaluated as

$$E_{\text{el}}^{\text{CG}}(d) = \sum_{i=1}^{N}\sum_{j=1}^{N} e_{\text{el}}(r_{\text{bead–bead}}) = \sum_{i=1}^{N}\sum_{j=1}^{N} \frac{Z_{\text{bead}}^2 e^2}{4\pi\varepsilon\varepsilon_0 \left|\mathbf{r}_i - \mathbf{r}_j\right|} e^{-\left|\mathbf{r}_i - \mathbf{r}_j\right|/b}. \tag{9}$$

Note that for the electrostatic interactions, the ligand length was taken into account in determining the interbead distance, depending on whether the beads were on the near, opposite, or same faces of the plates. Net van der Waals and electrostatic interactions were determined in this manner for a series of interparticle distances. The two potentials were then fit in Origin to curves of standard form for interparticle interactions:

$$E_{\text{vdW}}(d) = \frac{-A_1}{d^{B_1}} \tag{10}$$

and

$$E_{\text{el}}(d) = A_2 e^{-d/B_2} \tag{11}$$

with fitting parameters $A_1$, $A_2$, $B_1$, and $B_2$. The sum of these two curves, $E_{\text{tot}}$, was used to determine the position of the secondary energy minima, using Mathematica. We also evaluated the bead size effect by calculating the van der Waals interaction using a smaller bead size (8.75 nm in diameter) and the results are consistent with different bead sizes (Supplementary Fig. 3b).

To assess the validity of the CG approximation, we first compared the analytical and CG models assuming no rotation between the plate pair. A comparison of interaction curves based on the two calculations at various ionic strengths is given in Supplementary Fig. 3c. Supplementary Figure 3d demonstrates that the relevant features—secondary energy minima and their positions—are predicted with

reasonable agreement. We therefore proceeded to use the CG model to determine the interaction of plates with various degrees of rotation and/or truncation.

## Data availability

All the other remaining data are available within the article and Supplementary Files, or available from the corresponding authors upon request.

## Code availability

Custom Matlab codes for image processing, particle tracking, and particle interactions are available from the corresponding authors upon request.

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

## Acknowledgements

This work was supported by the National Science Foundation under Grant No. 1752517.

## Author contributions

B.L., A.K. and Q.C. designed the experiments. B.L., A.K., Z.W. and J.K. performed the experiments. B.L. and A.K. performed particle tracking and analysis of experimental data. J.W.S. and Z.O. contributed to the theoretical modeling of interparticle interaction. All contributed to the writing of the manuscript.

## Additional information

**Competing interests:** The authors declare no competing interests.

**Journal Peer Review Information:** *Nature Communications* thanks the anonymous reviewer(s) for their contribution to the peer review of this work. Peer reviewer reports are available.

