## [Peer Review File · Nature Communications]

Reviewers' comments:

Reviewer #1 (Remarks to the Author):

Chen and group has presented the work on "Hierarchical Self-Assembly of 3D Lattices from Polydisperse Anisometric Colloids". They have reported the hierarchical self-assembly of nano plates of silver and found the formation of meta-rod and subsequent honey-comb and hexagonal lattices. It is a well written manuscript and experiments were done carefully. However, similar work was published by many groups, for example, refs. [3,4,26,27], though the lattices formation is the new information in the present case. While the work presented is interesting to the materials community and extensive in nature, but is not rigorous enough to merit publication in Nature Communications in the current form. The novelty of the work is minimal in respect to the published work, for example ref. 3,4,26,27. Hence I would suggest that it is more suitable for the specialized journals of nanoscience or some materials science journal. Perhaps it may be reconsidered if they could prove the 3D lattices in real-space using some complementary techniques and include the full picture of DLVO calculations as mentioned in the comments section.

Major concerns:

- The authors claim that they form 3D structures with platelets which are not clear from the light microscopy. There is no characterization of 3D structures whatsoever.
- Page 3, authors mention that " The key difference is that, due to the larger plate basal plane, the meta-rods in this work have enhanced van der Waals attraction with the substrate, ", why would they get any van der Waals interactions between metal plate and substrate which here is a glass? I would expect gravity should pull it down.
- I don't think its a 3D structure anyway. More experimental evidence is needed.
- The DLVO calculations presented by the authors is quite simplified and over estimated. It doesn't include the retardation effects, for example, at 3.5 mM, the predicted DLVO energy is 1000 KT between triangular plates !! still they see rotational rearrangement Fig 4.
- It is surprised to see they still call it meta-rod !!. It is a flexible stack of plates.
- Is it specific to silver nanoplatelets or can be extended to other platelets?
- Fig.4, they found a sudden transition upon an ionic strength decrease from 6 mM to 2mM, from honey comb to hexagonal lattice. The authors still call it meta-rods instead of flexible chain. It may be due to the large fluctuations in the system.
- What is the error in the measurement of the platelet rotation?
- Are these silver nanoplates different from the nanoprisms reported by Mirkin's group?
- Fig. 1e seems to be still in the liquid state but authors claim it as a crystal.
- I feel the manuscript needs lots of improvements if it is to be reconsidered.

Minor comments

In many places, figure legends and the main text are clubbed together, difficult to read. Page numbers are not included in the manuscript which makes difficult to follow.

Reviewer #2 (Remarks to the Author):

Dear editor,

The authors report on the 3D assembly of inorganic platelets referred to as 'anisometric colloids, which integrate micrometer and nanometer dimensions into the same particle'. This is a fascinating area of colloid and materials sciences of steadily growing interest. Hierarchical materials with unprecedented

properties can be assembled from such building blocks and versatile synthesis routes are nowadays available to prepare anisometric colloids with reasonable yields and dispersity. The paper is interesting, well written, the experiments are performed and analysed in a thorough manner, the results are presented in a scholarly manner and discussed in depth. I do however disagree with a number of key statements, which is why I recommend to major revisions to address these aspects.

Major comments

- Key literature on the phase behaviour of inorganic platelets with high aspect ratio (as investigated herein) is not cited. The authors should cite major contributions to the field of various groups appropriately, including but not solely the well-known work of prof. Lekkerkerker. The statements of the authors that this class of colloids has remained 'underexplored' is only partially correct. Much more work has been done in this area for many years than suggested by the authors.
- The authors suggest that the hierarchical assembly of the polydisperse platelets is 'unexpected' (see for example page 1 Line 15, page 4 line 112), while in fact it is conform predictions by other groups (see for example van der Kooij, F. M.; Lekkerkerker, H. N. W. *Philosophical Transactions of the Royal Society of London Series a-Mathematical Physical and Engineering Sciences* 2001, 359 (1782), 985-995). The authors are perhaps unaware of the key studies on the so-called terminal polydispersity as they cite a very recent paper (ref 46, page 5, line 122), which is not the principle work on the terminal polydispersity of spheres, rods, nor platelets. It is well-known that suspensions of disperse hard-spheres (see e.g. for terminal dispersity of spheres Henderson, S. I.; Mortensen, T. C.; Underwood, S. M.; vanMegen, W., *Physica A* 1996, 233 (1-2), 102-116 & Pusey, P. N., *Journal De Physique* 1987, 48 (5), 709-712) are far less tolerant (as are rods) to dispersity than colloidal platelets.

Minor comments

- Page 2, line 45: "gravity [-] pose limitations which are in principle irrelevant here." Provide more concrete evidence for this statement, how small are the gravitational effects relative to the other interaction forces?
- Page 2, line 53: "size-focusing" Explain what is meant by this non-standard term.
- Page 3, line 71: Elaborate on the reproducibility of the pH and salt concentration in deionized water and 0.5mM NaCl. Under these conditions, pH fluctuations may be strong and would impact the phase behaviour as the platelets are electrostatically stabilized.
- Page 7, line 185: "made non-circular by changing the ionic strength" Reword for clarification
- Page 11, line 298: "transcend simulated triangular plates". Reword, unclear what is meant.

Reviewer #3 (Remarks to the Author):

This manuscript describes a joint experimental and theoretical study of the self-assembly of highly anisotropic colloids (thin plates) into superlattices. The plates spontaneously assemble into stacks (meta-rods) which then can be coerced to assemble into densely packed columnar lattices by simple densification via tilting. The authors show that ordered structures can be obtained in spite of the substantial polydispersity of plates. They furthermore demonstrate that the lattice symmetry can be manipulated via control of plate shape and ionic strength of the solvent. The authors show with experiments and calculations that the interactions between plates are well described as a competition between vdW attractions and electrostatic repulsions between charged ligands on the plate surface.

The manuscript describes an interesting colloidal system and the authors demonstrate exquisite control over the self-assembly behavior. The system has been well-characterized by multiple techniques and the theoretical analyzes of interactions is insightful and agree well with experiments. I recommend publication of the manuscript after the following points have been addressed.

1) The model presented in Fig. 1d suggests that the distance between stacked plates is about 100 nm. It would be highly desirable to experimentally verify this distance as a function of ionic strength and compare the results to the theory.

2) A related issue: The authors should determine the height distribution of meta-rods, i.e., how many plates typically stack up to form a rod.

3) It is unclear what exactly the authors mean by "size-focusing". If I understand correctly the authors claim that because each rod consists of several plates that can rotate, the size distribution of the projections of rods is narrower than the distribution of edge lengths of isolated plates. The authors claim that this effect enables assembly of ordered lattices but they present no evidence for this claim. Can they perform a control experiment with plates that do not stack up (e.g., because of larger thickness or larger diameter) and show that a monolayer of such plates does NOT form an ordered structure?

4) In Fig. 2c, the authors track the orientation of plates in two rods. How can the plate orientation be determined given that in Fig 2a the authors show that the projection of rods changes its shape on the second time scale?

5) Figure 3d: I do not fully understand the point of this graph. The authors change the shape of the plate (parameter m) and show that the triangular plates ($m=0$) interact more strongly than hexagonal plates ($m=1$). However, this effects seems to be mainly due to the fact that they are keeping $L+2L^*$ constant, which results in a larger area of the plate at $m=0$. If instead, the area of the plate is fixed, I assume they would not find a substantial difference between the interaction energies at $m=0$ and $m=1$. The authors should clarify why they choose to plot the energy at constant $L+2L^*$ and what the reader is supposed to learn from this plot.

6) Line 263. The authors claim that the honeycomb lattice transforms into a hexagonal lattice when the ionic strength is decreased. The only evidence for this claim comes from Supp Fig. 11, which in my opinion really only shows that the triangular lattice becomes disordered. In any case, the resulting structure is far inferior to the hexagonal lattice in Fig 1e. The authors should reconsider their claim of a structural transformation between two well-defined lattices.

7) Last sentence: "The real-time dynamics observed here can potentially extend to those systems." This analogy between COFs and the colloidal assembly discussed in the manuscript is very unclear. What exactly do the authors mean?

Reviewer 1

Reviewer 1 considered our work well-written, interesting and extensive in nature, with carefully performed experiments. We thank the reviewer for the comments, all of which we have addressed with new experiments, new analysis, and new calculations as detailed below, particularly on direct evidence of the three-dimensional (3D) structure and more thorough modeling of the interaction calculation.

Reviewer 1 Comment 1: *Similar work was published by many groups, for example, refs. [3,4,26,27], though the lattices formation is the new information in the present case.*

Reply: In the revised manuscript, we made it more clear that the novelty and focus of our work are the formation of the 3D lattice and the associated formation mechanism, including plate misalignment in the column and a narrowed building block size distribution during hierarchical assembly, as well as the control of the final lattice symmetry by changing the nanoscopic plate tip truncation and ionic strength. These aspects of the 3D lattice formation are the focus of all four main figures, which as the reviewer commented, have not been observed in assemblies from metallic plates.

In particular, the inter-column interaction controls critically the 3D lattice formation. The inter-column interaction profile is determined by the effective shape of the columns, which is contributed by the relative orientation of the plates. Meanwhile, details such as the plate tip morphology and the ionic strength together govern the relative orientation of the plates. These factors were not recognized previously and, more importantly, did not significantly modify the one-dimensional (1D) column assembly presented in previous work, such as those noted by the reviewer (refs. [3,4,26,27]).

Reviewer 1 Comments 2&4: *The authors claim that they form 3D structures with platelets which are not clear from the light microscopy. There is no characterization of 3D structures whatsoever. I don't think its a 3D structure anyway. More experimental evidence is needed.*

Reply: We thank the reviewer for helping us clarify our work better. In the original manuscript, we have provided the following experimental data to show the plates form into 3D structures.

1. We have shown first the formation of the 1D columns by (i) direct imaging of face-to-face stacking of individual plates into columns in the original Supplementary Fig. 2b (Supplementary Fig. 2c in the revised version), and (ii) direct imaging of a rotating column exhibiting different orientations in Fig. 1c, where we described, “The meta-rod geometry was verified under optical microscopy, as rotating meta-rods exhibit light, rectangular projections while lying on their sides and dark, polygonal projections while standing vertically on the substrate (Fig. 1c).”
2. Then the fact that the 1D columns serve as the building blocks of the lattice means that the lattice is 3D. This fact was supported by the polygonal projections of the standing columns shown in Fig. 1c, which are consistent with the building blocks in the final hexagonal lattice (Fig. 1e), and we observed the gradual increase of the concentration of standing columns leads to the 3D structure (Fig. 1f).
3. The projections of the lattice building blocks dynamically change, which is not possible if the lattice is formed from a two-dimensional (2D) layer of individual triangular plates. We stated in the original manuscript, “As shown in Fig. 2a and Supplementary Fig. 6, the constituent plates in a meta-rod rotate relative to each other (Supplementary Movie 2) so that the projections of meta-rods dynamically vary. The outlined contours of the meta-rod projections...fluctuate to average triangular plates of different orientations into a circular meta-rod projection as quantified in Fig. 2b.”

In the revised manuscript, we have made the 3D structure statement more clear by adding new data and direct 3D characterization as below:

1. Added Supplementary Movies 1, 2, and 4 to show the stacked-layer structure in each column: We added the movie of the 3D structure at intermediate column concentration (Supplementary Movie 4) corresponding to Fig. 1f. The empty space between each standing column makes it clear to see the plates stack upon each other into columns and exhibit fast rotations (see also Supplementary Fig. 6c, particularly the circled regions when the columns are slightly tilted). A similar aspect was emphasized by adding Supplementary Fig. 10 detailing different plate layers in columns comprising the 3D hexagonal lattice (presented in Fig. 2c, Supplementary Movie 6). We revised Supplementary Movie 5 to include views with and without contours overlaid, so that the column projection changes and “plates at the first few layers in the column are clearly identified based on their contrast difference under optical microscopy (Supplementary Movie 5)” as stated in the revised version. In addition, we have included optical microscopy movies (Supplementary Movies 1–2) corresponding to Supplementary Fig. 2c and main text Fig. 1c to show the process of plate stacking into columns and column rotations.
2. Scanning electron microscopy (SEM) characterization: We controllably dried the observed lattice after its formation in solution in the microscopy chamber, to maximally maintain the structure for SEM imaging as we monitored under the optical microscopy (although the plate-to-plate spacing in the same column disappears). In our newly added Supplementary Fig. 8, the SEM images clearly show that the plates are stacked face-to-face into columns, which further assemble laterally into 3D structures.

We have added a new Supplementary Note 5 to discuss all the above details for the characterization of 3D lattice structures in the revised version.

Reviewer 1 Comment 3: *Page 3, authors mention that “The key difference is that, due to the larger plate basal plane, the meta-rods in this work have enhanced van der Waals attraction with the substrate”, why would they get any van der Waals interactions between metal plate and substrate which here is a glass? I would expect gravity should pull it down.*

Reply: To elaborate our “enhanced van der Waals attraction with the substrate” discussion, in the revised manuscript, we added the calculation of the van der Waals attraction between a silver plate and the glass substrate across water. For two dissimilar objects, their van der Waals attraction across a medium can be computed following previous literature (Refs. 3–4 in SI) as detailed in Supplementary Note 3. The specific plate–substrate interaction can be calculated following previous analytical models (Ref. 3 in SI). Thus we obtained a plate–substrate van der Waals attraction of about $-18 k_B T$ at a plate–substrate separation of 100 nm and still a few $k_B T$ at ~ 250 nm plate–substrate separation, strong enough to keep the silver plates lying flat and support the columns “standing” on the substrate. We agree with the reviewer that gravitational force can also contribute to the lying flat configuration of the columns ($\sim 2.5 k_B T$) and have now added that discussion in Supplementary Note 3.

Reviewer 1 Comment 5: *The DLVO calculations presented by the authors is quite simplified and overestimated. It doesn't include the retardation effects, for example, at 3.5 mM, the predicted DLVO energy is 1000 kT between triangular plates !! still they see rotational rearrangement Fig 4.*

Reply: We thank the reviewer for this great point about retardation effects in the van der Waals attraction calculation! Retardation effects were not included in our original calculation because retardation effects are usually negligible in nanoparticle assembly. But we agree with the reviewer to consider the retardation effect since for our anisometric system, the adjacent plates in an assembled column can be more than 100

nm apart, making the retardation effects matter. In the revised manuscript, we have included the retardation effects of van der Waals attraction, following previous literature (Refs. 41, 59–61 in the main text), in both the analytical and discrete model calculations. We updated the calculations displayed in Fig. 1d, Fig. 3b–e, Fig. 4e, Supplementary Fig. 3, and Supplementary Table 3. All the details of the revised calculation have been added to the Methods section.

Note that such modifications do not alter any scientific conclusions made in the original manuscript, such as the effects of tip truncation and ionic strength on determining plate relative rotation and the effective column shape. Including the retardation effects decreases the total interaction energy by a factor of 4–6 at most (or by a smaller factor at high ionic strength when plates are closer with a smaller retardation), but otherwise the results remain qualitatively the same.

We also note that the rotational rearrangement is not determined by the absolute interaction strength, but by the interaction energy difference during the relative rotation of two adjacent plates, which we estimated based on the Boltzmann distribution (relative probability, $p_{\text{rel}}(\Delta\theta) \approx e^{-\Delta E/k_{\text{B}}T}$, where ΔE is the difference of E_{tot} for two plates at different relative orientations). As a result, for the hexagonal lattice, relative plate rotation is allowed due to a small ΔE (Fig. 3c), even though the absolute interaction strength is a few $k_{\text{B}}T$.

Reviewer 1 Comment 6: *It is surprised to see they still call it meta-rod!! It is a flexible stack of plates.*

Reply: We thank the reviewer for this detailed comment. We agree that the meta-rod terminology is non-standard, and we now refer to the stacked plates as “columns”.

Reviewer 1 Comment 7: *Is it specific to silver nanoplatelets or can be extended to other platelets?*

Reply: We thank the reviewer for this constructive comment. We expect the 3D structure formation can be extended to other plate systems for the following reasons.

1. Our observed 3D structure is formed due to a hierarchical assembly driven by directional interplate interaction, coupled with the control over the effective secondary building block geometry. These key components could in principle be achieved in other platelet systems with an appropriate balance of attractive and repulsive interactions. This is consistent with our new experimental data shown in Supplementary Fig. 7, where such ordered 3D hexagonal lattices were robustly observed in multiple plate samples of different size and tip truncation.
2. As the reviewer recognized, our work is the first reporting the 3D hexagonal lattice from metallic plates, while only 1D structures are observed in previous studies. Yet, such 3D hexagonal lattices have been reported based on small angle X-ray scattering (SAXS) characterizations for *hexagonal* nanoplates of other compositions such as $\text{Al}(\text{OH})_3$ (Refs. 26, 27, 29 in the main text) and $\text{Ni}(\text{OH})_2$ nanoplates (Ref. 28 in the main text). In these previous studies, the ensemble structure was characterized, but because the plates are small (below 200 nm in side length), the real-time and real-space dynamics we reported here for our anisometric plates with micrometer side length (Figs. 1–4) have not been previously reported. Direct imaging and understanding of the lattice formation mechanism, plate stacking into column, 3D lattice relaxation, plate rotation as well as the plate tip truncation and their impact on the effective column shape have not been studied.

Reviewer 1 Comment 8: *Fig.4, they found a sudden transition upon an ionic strength decrease from 6 mM to 2mM, from honey comb to hexagonal lattice. The authors still call it meta-rods instead of flexible chain. It may be due to the large fluctuations in the system.*

Reply: In the revised manuscript, we note that the honeycomb lattice domains at 6 mM (Supplementary Fig. 15) are stable for up to days when the ionic strength remains unchanged. It is only when a decrease in

ionic strength from 6 mM to 2 mM reduces the interaction energy penalty for relative plate rotation (as discussed in Fig. 3b–c) that there are large fluctuations leading to conversion to a hexagonal lattice. As a result, we observed a transition from the space-filling honeycomb lattice composed of triangular columns to a relatively loosely packed structure which exhibited hexagonal packing.

Reviewer 1 Comment 9: *What is the error in the measurement of the platelet rotation?*

Reply: In the revised manuscript, we included the measurement errors in the plate rotation for Fig. 2c ($\pm 5^\circ$) and Fig. 4f ($\pm 2^\circ$).

Reviewer 1 Comment 10: *Are these silver nanoplates different from the nanoprisms reported by Mirkin's group?*

Reply: The silver plates used in our work and the typical nanoprisms reported by Mirkin's group (Au, Ag, Ag-Ag₂S as shown in Young, *Proc. Natl. Acad. Sci. U. S. A.*, 2012; Métraux, *Adv. Mater.*, 2005; Shahjamali, *ACS Nano*, 2016) have both similarities and differences. Both can be rendered to exhibit the same type of interparticle interactions (van der Waals, electrostatic) if coated with similarly charged ligands. We expect Mirkin's nanoprisms can also form into the 3D lattices we observed here, if the net prism–prism interaction is carefully tuned to favor relative rotations of adjacent prisms in the same column, and if there is a flat substrate to support vertically standing columns to laterally interact. However, Mirkin's nanoprisms (typically with side lengths of 30–300 nm) are smaller than the silver plates we use, which have micrometer side lengths. This micrometer side length gives rise to high aspect ratio (~66) and well-exhibited, tunable tip geometry. It also serves as the key feature to allow direct imaging of solution phase behaviors by optical microscopy in real time and in real space. The tunable tip geometry determines the interparticle energy profile during plate relative rotation, the effective shape of the columns, and eventually the final 3D lattice structures.

Reviewer 1 Comment 11: *Fig. 1e seems to be still in the liquid state but authors claim it as a crystal.*

Reply: In the original manuscript, we followed the colloidal crystal reference (Ref. 52 in the original text) to calculate the six-fold local bond orientation order parameter $|\psi_{6j}|$ to quantify the crystallinity of the hexagonal lattice. In the revised version, we have added additional colloidal crystal references, which define a domain as crystal when the average $|\psi_{6j}|$ over all the building blocks is over 0.8 (Ref. 46 in the main text), and define particles with $|\psi_{6j}|$ larger than 0.5–0.8 as in the crystalline phase (Refs. 46–47 in the main text). Consistently, our analysis shows that 83% columns in the hexagonal lattice have $|\psi_{6j}|$ larger than 0.8, and the average $|\psi_{6j}|$ over all the columns is 0.87, both of which agree with highly crystalline structures following literature definition.

Reviewer 1 Comment 13: *In many places, figure legends and the main text are clubbed together, difficult to read. Page numbers are not included in the manuscript which makes difficult to follow.*

Reply: Thanks! These details are all fixed!

Reviewer 2

Reviewer 2 considers our anisometric colloidal building blocks as a fascinating area of colloid and materials sciences of steadily growing interest. The reviewer also considers our manuscript interesting and well-written, the experiments performed and analyzed in a thorough manner, and the results presented and discussed in depth. The reviewer recommended us to clarify a few statements, and to include a few more references in our manuscript, which we have all addressed in our revised manuscript.

Reviewer 2 Comment 1: *Key literature on the phase behaviour of inorganic platelets with high aspect ratio (as investigated herein) is not cited. The authors should cite major contributions to the field of various groups appropriately, including but not solely the well-known work of prof. Lekkerkerker. The statements of the authors that this class of colloids has remained ‘underexplored’ is only partially correct. Much more work has been done in this area for many years than suggested by the authors.*

Reply: We thank the reviewer for the great and constructive suggestion! The silver plates we studied have micron-scale side length and nanoscale thickness (aspect ratio ~66), different from the inorganic platelets studied previously with nanoscale side length and aspect ratio typically < 15. We completely agree with the reviewer that the previous studies are inspirational. In the revised manuscript, we cited more references of the previous work and also noted the novelty of our system. Our work is the first reporting 3D hexagonal lattice from metallic plates, where only one-dimensional (1D) structures have been observed. In the previous studies of 3D hexagonal lattices from *hexagonal* inorganic nanoplates of other compositions, the ensemble structure was characterized, but because the plates are small (below 200 nm in side length), the real-time and real-space dynamics we reported here for our anisometric plates with micrometer side length (Figs. 1–4) have not been previously reported. Direct imaging of lattice formation process, plate stacking into column, 3D lattice relaxation, plate rotation as well as the plate tip truncation and their impact on the effective column shape have not been studied. These aspects of direct imaging allowed us to provide an in-depth explanation of why and how 3D lattices can form from such particles.

Our added references include the $\text{Al}(\text{OH})_3$ plates with a diameter of ~200 nm and aspect ratio of ~15 from Prof. Lekkerkerker’s group (Refs. 26, 27, 29 in the main text), $\text{Ni}(\text{OH})_2$ plates with a diameter of ~100 nm and aspect ratio of ~5 from Prof. Rennie’s group (Ref. 28 in the main text), CuS nanoplates with a diameter of ~15 nm and aspect ratio of ~3 from Prof. Korgel’s group (Ref. 30 in the main text), Cu_2S nanoplates with a diameter of ~18 nm and aspect ratio of ~4 from Prof. Chen’s group (Ref. 31 in the main text), and simulation studies from Prof. Frenkel’s group (Ref. 54 in the main text) and Prof. van Duijneveldt’s group (Ref. 55 in the main text).

Reviewer 2 Comment 2: *The authors suggest that the hierarchical assembly of the polydisperse platelets is ‘unexpected’ (see for example page 1 Line 15, page 4 line 112), while in fact it is conform predictions by other groups (see for example van der Kooij, F. M.; Lekkerkerker, H. N. W. *Philosophical Transactions of the Royal Society of London Series a-Mathematical Physical and Engineering Sciences* 2001, 359 (1782), 985-995). The authors are perhaps unaware of the key studies on the so-called terminal polydispersity as they cite a very recent paper (ref 46, page 5, line 122), which is not the principle work on the terminal polydispersity of spheres, rods, nor platelets. It is well-known that suspensions of disperse hard-spheres (see e.g. for terminal dispersity of spheres Henderson, S. I.; Mortensen, T. C.; Underwood, S. M.; vanMegen, W., *Physica A* 1996, 233 (1-2), 102-116 & Pusey, P. N., *Journal De Physique* 1987, 48 (5), 709-712) are far less tolerant (as are rods) to dispersity than colloidal platelets.*

Reply: We thank the reviewer for making our work more comprehensive! In the revised manuscript, we have clarified that “unexpected” is in the context of 3D lattices from metallic plates, as only 1D structures are observed in previous studies of metallic plates (Refs. 3, 4, 26, 27 in the original text as pointed out by Reviewer 1). We have included more references (Refs. 26, 27, 29 in the main text) on the polydispersity of

plates including those suggested by the reviewer. We also revised our discussion on the plate polydispersity by emphasizing on the mechanism: a greatly reduced polydispersity from plate stacking into columns as evidenced from direct optical microscopy imaging. This mechanism was not elucidated in previous work, which we state now clearly in the manuscript: “However, the secondary building blocks of columns average their sizes due to plate stacking, thereby rendering them greatly reduced polydispersity (8%) to assemble into ordered lattice (Fig. 1b). Such a size-focusing effect can potentially account for the formation of similar 3D lattices from polydisperse Al(OH)₃ nanoplates presented in earlier work, where ensemble structure was characterized by SAXS^{26,27}.”

Reviewer 2 Comment 3: *Page 2, line 45: “gravity [-] pose limitations which are in principle irrelevant here.” Provide more concrete evidence for this statement, how small are the gravitational effects relative to the other interaction forces?*

Reply: The fluctuation height of colloids determines the extent of 3D assembled structure the colloids can form into (Ref. 39 in the main text). In the original manuscript, we supported this general statement with the calculated fluctuation height (defined as the height colloids can thermally fluctuate up to against gravity in solvent) of our plates, and stated, “The plates exhibit vivid in-plane and out-of-plane rotations in 3D as observed under optical microscopy, with vertical fluctuation height⁴² as much as 11 times the plate thickness (Supplementary Fig. 2a, Supplementary Note 2). Such out-of-plane rotation in 3D, given one order of magnitude difference between the densities of silver and water, is possible due to the nanoscale thickness.”

In the revised version, we have included additional calculations on the vertical distribution of plates based on the Boltzmann distribution in Supplementary Note 1 and Supplementary Fig. 2a. The calculation shows that the plates can go up to micrometers to allow assembly in 3D.

Reviewer 2 Comment 4: *Page 2, line 53: “size-focusing” Explain what is meant by this non-standard term.*

Reply: We thank the reviewer for this detailed suggestion. In the revised version, we made it more clear that “size-focusing” means an effective narrowing of the building block size distribution, from 16% polydispersity in the plates to 8% polydispersity in the columns, i.e. the secondary building blocks (Fig. 1b). This effect is due to the plate stacking into columns, which renders their polydispersity reduced enough to assemble into ordered lattices.

Reviewer 2 Comment 5: *Page 3, line 71: Elaborate on the reproducibility of the pH and salt concentration in deionized water and 0.5mM NaCl. Under these conditions, pH fluctuations may be strong and would impact the phase behaviour as the platelets are electrostatically stabilized.*

Reply: We thank the reviewer for this great point! In the revised manuscript, we added discussions on the effects of pH and ionic strength fluctuations due to CO₂ dissolved in water. As detailed from calculations shown in Supplementary Note 2, the solution pH after considering CO₂ dissolving in water is 5.7, which still keeps 99% of the charged ligands fully deprotonated and does not change the deprotonation of ligands, namely the surface charge density of the plates. The ionic strength increase after considering CO₂ dissolving in water is 0.002 mM, only 0.4% of the ionic strength of the solution at 0.5 mM NaCl condition, which is also negligible.

Reviewer 2 Comment 6: *i) Page 7, line 185: “made non-circular by changing the ionic strength” Reword for clarification. ii) Page 11, line 298: “transcend simulated triangular plates”. Reword, unclear what is meant.*

Reply: Thanks! Both are fixed now:

- i. the effective shape of columns can be controlled by changing the ionic strength;
- ii. our experimentally observed 3D hierarchical lattices are different from those adopted by simulated triangular prisms, which only form into honeycomb lattices due to hard-core exclusion.

To Reviewer 3

Reviewer 3 considers that our work describes an interesting colloidal system and demonstrates exquisite control over the self-assembly behavior. The reviewer also considers that our system has been well-characterized by multiple techniques; the theoretical analysis of interactions is insightful and agrees well with experiments. The reviewer recommends publication of the manuscript upon the following comments, which we have all addressed in the revised manuscript.

Reviewer 3 Comment 1: *The model presented in Fig. 1d suggests that the distance between stacked plates is about 100 nm. It would be highly desirable to experimentally verify this distance as a function of ionic strength and compare the results to the theory.*

Reply: We thank the reviewer for the great suggestion! In our earlier work on the columns formed from similarly stacked plates (Ref. 2 in the main text), we have measured the center-to-center distance using small-angle X-ray scattering (SAXS) and the measured values were about 117 nm at 1.4 mM ionic strength and 86 nm at 3.0 mM ionic strength. These results were also consistent with theoretical predictions. In the revised version, we have performed SAXS measurement to the major plate sample used in this work (Figs. 1–2) and the measured d spacing still agrees with that predicted by the analytical and decently well with that calculated from discretized models (Supplementary Fig. 4, Supplementary Table 3).

Reviewer 3 Comment 2: *The authors should determine the height distribution of meta-rods, i.e., how many plates typically stack up to form a rod.*

Reply: We thank the reviewer for the great suggestion! We have added two graphs in the revised version. One is on the length distribution of columns (L_c) measured from optical microscopy movies (with error bars noted) when columns exhibit out-of-plane rotations (Supplementary Fig. 2d). Interestingly, as shown in the other graph, the distribution of column aspect ratio (defined as the ratio of its length L_c to its circumscribed diameter D , Supplementary Fig. 2e) fluctuates around 1.0 ± 0.1 , which means one column is composed of ~15 plates estimated from the center-to-center spacing between adjacent plates in the column.

Reviewer 3 Comment 3: *It is unclear what exactly the authors mean by “size-focusing”. If I understand correctly the authors claim that because each rod consists of several plates that can rotate, the size distribution of the projections of rods is narrower than the distribution of edge lengths of isolated plates. The authors claim that this effect enables assembly of ordered lattices but they present no evidence for this claim. Can they perform a control experiment with plates that do not stack up (e.g., because of larger thickness or larger diameter) and show that a monolayer of such plates does NOT form an ordered structure?*

Reply: In the revised manuscript, we made it more clear that “size-focusing”, as the reviewer pointed out, means an effective narrowing of the building block size distribution, from 16% polydispersity in the plates to 8% polydispersity in the columns, i.e. the secondary building blocks (Fig. 1b). This effect is due to the plate stacking into columns, which render reduced polydispersity to assemble into ordered lattices. We also added new experiment on the 2D monolayer assembly from plates (Supplementary Fig. 16) following the reviewer’s suggestion and stated, “Control experiments on the 2D assembly of silver plates at different plate concentrations did not show ordered structure formation (Supplementary Fig. 16).” This is because in the 2D assembly, plates themselves are the building blocks, whose polydispersity is too large to crystallize.

Reviewer 3 Comment 4: *i) In Fig. 2c, the authors track the orientation of plates in two rods. How can the plate orientation be determined given that in Fig 2a the authors show that the projection of rods changes its shape on the second time scale?*

Reply: We thank the reviewer for reading our paper carefully! Fig. 2a and Fig. 2c correspond to different concentrations of columns (isolated *vs* closely packed). We used a greater intensity of illumination in Fig. 2c to image the closely packed lattice structures, which gives a better contrast of plates at the first layer of columns, decent enough to be tracked under optical microscopy, as shown in Supplementary Movie 6. In the revised Fig. 2c caption, we have also included measurement error ($\pm 5^\circ$).

As to the time scale, we tracked the orientations of the plates at the first layer of the neighboring column pair, and the videos were recorded at a rate of 17.2 frames per second. In comparison, the projections of columns change over time on the time scale of seconds. Both the spatial and temporal resolutions are sufficient for us to track the plate orientations in Fig. 2c.

Reviewer 3 Comment 5: *Figure 3d: I do not fully understand the point of this graph. The authors change the shape of the plate (parameter m) and show that the triangular plates ($m=0$) interact more strongly than hexagonal plates ($m=1$). However, this effects seems to be mainly due to the fact that they are keeping $L+2L'$ constant, which results in a larger area of the plate at $m=0$. If instead, the area of the plate is fixed, I assume they would not find a substantial difference between the interaction energies at $m=0$ and $m=1$. The authors should clarify why they choose to plot the energy at constant $L+2L'$ and what the reader is supposed to learn from this plot.*

Reply: We thank the reviewer for the great comment and suggestion! In the original manuscript, we started our calculation from a perfect triangular plate ($m = 0$) and then directly introduced truncation on the same plate, which makes their total side length ($L + 2L'$) a constant. We completely agree with the reviewer's comment. In the revised Fig. 3d–e, we presented the energy diagrams with the basal plane area fixed for differently truncated plates. Although the net interaction minimum at $\Delta\theta = 0$ is mostly independent of the extent of tip truncation now that the area is fixed, our statement on how the plate tip truncation affects the allowed *relative* rotation of the plates still holds.

Reviewer 3 Comment 6: *Line 263. The authors claim that the honeycomb lattice transforms into a hexagonal lattice when the ionic strength is decreased. The only evidence for this claim comes from Supp Fig. 11, which in my opinion really only shows that the triangular lattice becomes disordered. In any case, the resulting structure is far inferior to the hexagonal lattice in Fig 1e. The authors should reconsider their claim of a structural transformation between two well-defined lattices.*

Reply: We appreciate this great suggestion! Originally the hexagonal lattice was inferred from the six-fold symmetry and the increased distance between nearest column pairs, both of which are indicated by the fast Fourier transform (the original Supplementary Fig. 11, Supplementary Fig. 15 in the revised). In the revised manuscript, we have changed our statement as follows: “Furthermore, upon an ionic strength decrease from 6 mM to 2 mM in this system, we observed a transition from the space-filling honeycomb lattice composed of triangular columns to a relatively loosely packed structure which exhibits hexagonal packing due to increased electrostatic repulsion and lowered penalty for relative plate rotation (Supplementary Fig. 15).”

Reviewer 3 Comment 7: *Last sentence: “The real-time dynamics observed here can potentially extend to those systems.” This analogy between COFs and the colloidal assembly discussed in the manuscript is very unclear. What exactly do the authors mean?*

Reply: We thank the reviewer for this thoughtful comment! In the revised version, we added more details for the similarity between the formation of our crystals and the formation of covalent organic frameworks: “Molecules such as boron-containing “connectors” or hydrocarbon “linkers” can form into covalent organic frameworks, with cage architecture and rotational order developed upon a delicate balance between intermolecular attraction (van der Waals, π - π stacking) and electrostatic repulsion⁵⁷. The formation of

covalent organic frameworks is similar to the formation of our honeycomb lattice from triangular plates, upon a balance of interparticle interactions.”

REVIEWERS' COMMENTS:

Reviewer #1 (Remarks to the Author):

The authors have revised the manuscript extensively and satisfactorily answered all my queries. However, there are still few issues to be clarified before publication of the manuscript in Nature Communications.

1. It is not clear how do authors compare/correlate the results of SEM images obtained from drying with the solution state structures, to confirm the 3D structures. Drying forces can assemble them into ordered stacks. Authors should clarify this.
2. The hexagonal lattice showed in Fig. 4d is not clear, it may be magnified to get a clear picture.
3. FTT patterns in Fig. 1e and Fig. 4d is quite hazy. Try to get a clear pattern like the one shown in Fig.7 (Supplementary Information).

Reviewer #3 (Remarks to the Author):

The authors have put significant effort into revising their manuscript according to all reviewers' suggestions. My concerns and comments have all been addressed. I recommend publication of the manuscript.

Reviewer 1

Reviewer 1 commented that our revisions are extensive and have satisfactorily answered the queries. The reviewer suggests clarifying a few issues before publication on Nature Communications, which we have all addressed.

Reviewer 1 Comment 1: *It is not clear how do authors compare/correlate the results of SEM images obtained from drying with the solution state structures, to confirm the 3D structures. Drying forces can assemble them into ordered stacks. Authors should clarify this.*

Reply: We have now added further clarification in Supplementary Note 5 and the main text that the SEM sample was prepared directly from the assembled lattices formed at 3 mM ionic strength in an optical microscopy chamber (following the procedures we used in optical microscopy imaging of the lattices). As the 3D lattices are closely packed, gentle drying does not lead to significant distortion or collapsing of the structures, which we confirmed from the SEM images and serves as the evidence of the 3D nature of the assembled lattices.

Reviewer 1 Comments 2: *The hexagonal lattice showed in Fig. 4d is not clear, it may be magnified to get a clear picture. FTT patterns in Fig. 1e and Fig. 4d is quite hazy. Try to get a clear pattern like the one shown in Fig.7 (Supplementary Information).*

Reply: We thank the reviewer for the great suggestions! Both are now addressed.

Reviewer 3

Reviewer 3 states that we have put significant effort into revising our manuscript according to all reviewers' suggestions, and the reviewer's concerns and comments have all been addressed. The reviewer recommends publication of the manuscript.